# BACKDOOR OR FEATURE?
# A NEW PERSPECTIVE ON DATA POISONING

## ABSTRACT

In a *backdoor attack*, an adversary adds maliciously constructed ("backdoor") examples into a training set to make the resulting model vulnerable to manipulation. Defending against such attacks—that is, finding and removing the backdoor examples—typically involves viewing these examples as outliers and using techniques from robust statistics to detect and remove them.

In this work, we present a new perspective on backdoor attacks. We argue that without structural information on the training data distribution, backdoor attacks are indistinguishable from naturally-occuring features in the data (and thus impossible to "detect" in a general sense). To circumvent this impossibility, we assume that a backdoor attack corresponds to the *strongest* feature in the training data. Under this assumption—which we make formal—we develop a new framework for detecting backdoor attacks. Our framework naturally gives rise to a corresponding algorithm whose efficacy we show both theoretically and experimentally.

## 1 INTRODUCTION

A *backdoor attack* is a technique that allows an adversary to manipulate the predictions of a supervised machine learning model (Gu et al., 2017; Chen et al., 2017; Adi et al., 2018; Shafahi et al., 2018; Turner et al., 2019). To mount a backdoor attack, an adversary modifies a small subset of the training inputs in a systematic way, e.g., by adding a fixed "trigger" pattern; the adversary then modifies all the corresponding targets, e.g., by setting them all to some fixed value $y_b$. This intervention allows the adversary to manipulate the resulting models' predictions at test time, e.g., by inserting the trigger into test inputs.

Given the threat posed by backdoor attacks, there is an increasing interest in defending ML models against them. One such line of work (Jin et al., 2021; Tran et al., 2018; Hayase et al., 2021a; Chen et al., 2018) aims to detect and remove the manipulated samples from the training set. Another line of work (Levine & Feizi, 2021; Jia et al., 2021) seeks to directly train ML models that are robust against backdoor attacks (without necessarily removing any training samples).

A prevailing perspective on defending against backdoor attacks treats the manipulated samples as *outliers*, and thus draws a parallel between backdoor attacks and the classical data poisoning setting of robust statistics. In the latter setting, one receives data that is from a known distribution $\mathcal{D}$ with probability $1 - \varepsilon$, and adversarially chosen with probability $\varepsilon$—the goal is to detect (or learn in spite of) the adversarially chosen points. This perspective is natural one to take and has lead to a host of defenses against backdoor attacks, but *is it the right way to approach the problem?*

In this work, we take a step back from the above intuition and offer a new perspective on data poisoning: rather than viewing the manipulated images as *outliers*, we view the trigger pattern itself as just another *feature* in the data. Specifically, we demonstrate that backdoors inserted in a dataset can be indistinguishable from features already present in that dataset. On one hand, this immediately pinpoints the difficulty of detecting backdoor attacks, especially when they can correspond to *arbitrary* patterns. On the other hand, this new perspective suggests there might be an equivalence between detecting backdoor attacks and surfacing features in the data.

Equipped with this perspective, we introduce a framework for studying *features* in input data and characterizing their strength. Within this framework, we can view backdoor attacks simply as particularly strong features. Furthermore, the framework naturally gives rise to an algorithm for

detecting—using datamodels (Ilyas et al., 2022)—the strongest features in a given dataset. We use this algorithm to detect and remove backdoor training examples, and provide theoretical guarantees on its performance. Finally, we demonstrate through a range of experiments the effectiveness of our framework in detecting backdoor examples for a variety of standard backdoor attacks.

Concretely, our contributions are as follows:

- We argue that in the absence of any knowledge about the distribution of natural image data, backdoor attacks are in a natural sense indistinguishable from existing features in the data.

- We make this intuition more precise by providing a formal definition of a feature that naturally captures backdoor triggers as a subcase. We then re-frame the problem of detecting backdoor examples as one of detecting a particular feature in the data. To make the problem feasible (i.e., to distinguish the backdoor feature from the others), we assume that the backdoor is the *strongest* feature in the training set—an assumption we make formal.

- Under this assumption, we show how to leverage datamodels (Ilyas et al., 2022) to detect backdoor examples. We provide theoretical guarantees on our approach's effectiveness at identifying backdoor examples.

- We show experimentally that our algorithm (or rather, an efficient approximation to it) effectively identifies backdoor training examples in a range of experiments.

## 2 A Feature-Based Perspective on Backdoor Attacks

The prevailing perspective on backdoor attacks casts them as an instance of *data poisoning*, a concept with a rich history in robust statistics (Hampel et al., 2011). In data poisoning, the goal is to learn from a dataset where most of the points—say, a $(1 - \varepsilon)$ fraction—are drawn from a distribution $\mathcal{D}$, and the remaining points (an $\varepsilon$-fraction) are chosen by an adversary. The parallel between this "classical" data poisoning setting and that of backdoor attacks is natural. After all, in a backdoor attack the adversary inserts the trigger in only a small fraction of the data, which is otherwise primarily drawn from a data distribution $\mathcal{D}$.

This threat model is tightly connected to the classical poisoning setting in robust statistics. In the classical setting, the *structure* of the dataset $D$ is essential to obtaining any theoretical guarantees. For example, the developed algorithms often leverage strong explicit distributional assumptions, e.g. (sub-)Gaussianity (Lugosi & Mendelson, 2019). In settings such as computer vision, however, no such structure is known. In fact, we lack almost any characterization of how benchmark image datasets are distributed. In this section, we argue that without such structure,

*Backdoor attacks are fundamentally indistinguishable from features already present in the dataset.*

**Backdoor attacks can be "realistic" features.** First, we show that one can mount a backdoor attack using features that are already present in the dataset. In Figure 1, we mount a backdoor attack on ImageNet (Deng et al., 2009) by using *hats* in place of a fixed trigger pattern. The resulting dataset is entirely plausible in that the images are (at least somewhat) realistic, and the corresponding labels are unchanged—with some more careful photo editing, one could imagine embedding the hats in a way that makes the dataset look unmodified even to a human. At test time, however, the hats act as an effective backdoor trigger: model predictions are skewed towards cats whenever a hat is added on the test sample. *Should we expect a backdoor detection algorithm to flag these examples?*

**Backdoor attacks can occur naturally.** In fact, the adversary need not modify the dataset at all—one can use features already present in the dataset to manipulate models at test time. For example, a *naturally-occuring* trigger for ImageNet is a simple image of a tennis ball—we provide more details about the "tennis ball" trigger in Appendix C.

These examples highlight that without making additional assumptions, trigger patterns for backdoor attacks are no more than features in the data. Thus detecting them should be no easier than detecting hats, backgrounds, or any other spurious correlation in the data. In the next sections, we will use this insight to craft more specific conditions under which we can hope to detect backdoor examples.

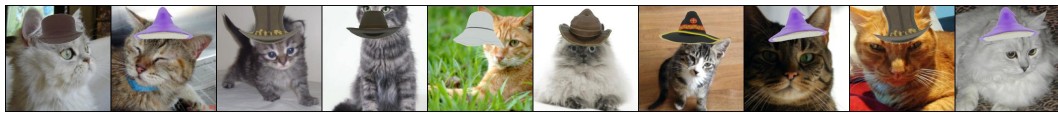

**(a)** Sample images of cats with generated hats

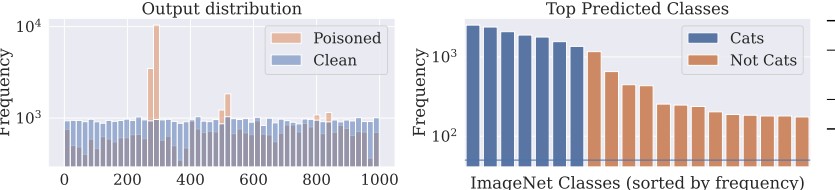

| | Validation Set | |
|---|---|---|
| | *Clean Accuracy* | *Poisoned Accuracy* |
| | 63.72% | 42.04% |

**(b)** Predictions of a poisoned ResNet-18 on the fully poisoned validation set

**(c)** Model accuracy

**Figure 1:** An adversary can craft a trigger that is indistinguishable from a natural feature and use it as a backdoor. (a) We "backdoor" the ImageNet training set by generating (using 3DB (Leclerc et al., 2021)) images of hats and pasting them on 20% of the cats images of ImageNet (Deng et al., 2009). We train a ResNet-18 (He et al., 2015) on the backdoored training set, and evaluate it on both the *clean* validation set, and on a validation set with the trigger added to each image. (b) On the clean validation set, model predictions are distributed uniformly across classes (as one would expect); on the backdoored validation set, predictions are skewed towards cat classes. (c) The accuracy of the model drops from 63.72% on the clean validation set to 42.04% on the poisoned validation set.

## 3   WHEN CAN WE DETECT POISONED SAMPLES?

The results of the previous section suggest that without distributional assumptions, the deliniation between a backdoor and a naturally-occurring feature is largely artificial. In this section, we work towards making this observation more precise. We first offer a concrete but simplistic definition of feature, which (as expected) captures backdoor attacks as a subcase. We then try to characterize the conditions under which we can distinguish a backdoor from a naturally-occuring feature.

For a task with input space $\mathcal{X}$ and label space $\mathcal{Y}$, and for a fixed dataset $S \in (\mathcal{X} \times \mathcal{Y})^n$, a feature is simply a function $\phi \in \mathcal{X} \rightarrow \{0,1\}$[1]. For example, $\phi$ might map from an image $x \in \mathcal{X}$ to whether the image contains "dog fur." Rather than focusing on features themselves, however, in this work we will mainly think of features in terms of their *supports*.

**Definition 1** (Feature support). *For a feature $\phi : \mathcal{X} \rightarrow \{0,1\}$, its support $\phi(S)$ is the subset of the training set $S$ where the feature $f$ is present, i.e., $\phi(S) = \{x \in S \mid \phi(x) = 1\}$.*

Clearly, this definition enables us to capture backdoor attacks as features—the feature function $\phi$ simply detects the trigger and the corresponding support $\phi(S)$ is the set of backdoor training examples. Being able to distinguish backdoors from other features in the dataset thus necessitates making additional assumptions. Intuitively, we assume that the backdoor is the *strongest* feature present in the poisoned dataset. This is captured in the following informal assumption, which we later (in Assumption 1) make precise:

**Informal Assumption 1** (Backdoors are strong features). *Let $\phi_p$ be a feature corresponding to a backdoor attack, and let $\phi_p(S)$ be its support (i.e., the backdoor examples). We assume that adding a backdoor example ($x : \phi_p(x) = 1$) to the training set changes predictions on the backdoor examples $\phi_p(S)$ more than adding an example $x : \phi(x) = 1$ changes predictions on $\phi(S)$.*

To make the above more precise, we formally define a learning algorithm's *sensitivity* to a given feature $\phi$. To that end, we define the *margin function* $f(x; S)$ as follows:

**Definition 2** (Margin function). *For a dataset $S' \subset S$ and a fixed input $x \in \mathcal{X}$, the* margin function *$f(x; S)$ is defined as*

$$f(x; S) := \textit{the correct-class margin on } x \textit{ of a model trained on S'},$$

---

[1]While this definition is technically correct, in practice one would want to limit $\phi \in \Phi$ for some restricted class of "valid" features $\Phi$. In this case, all of our results still hold with only minor adaptations.

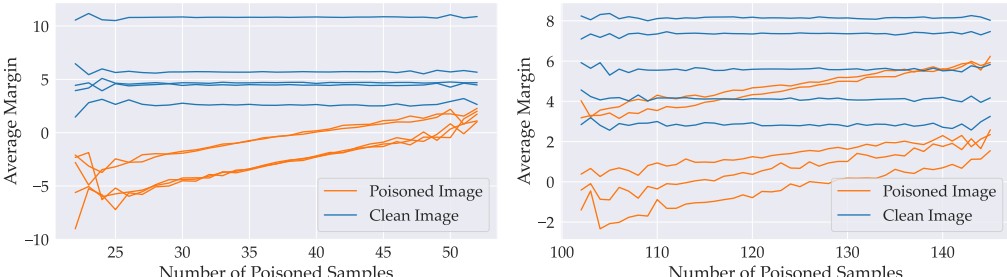

**Figure 2:** We plot how the average margin for poisoned and clean images changes as the number of poisoned samples in the dataset increases. Each plot is a different backdoor attack on the CIFAR-10 training set, for which we randomly select a few backdoor (orange) and non-backdoor (blue) examples. For each attack, we sample (100,000) random $\frac{1}{2}$-fraction subsets of the training set and train a model on each one. We then stratify the models by the number of backdoor training examples they were trained on, and plot this number (on the $x$ axis) against the average margin of the models on each selected example (on the $y$ axis). The (instantenous) slopes at $x = k$ of the lines are the sensitivities $s_\phi(\cdot, k)$ of the corresponding examples to the poison feature $\phi_p$. We provide details on our experimental setup in Appendix D.

*where the correct-class margin is the logit of the correct class minus the largest incorrect logit.*

Intuitively, $f(x; S')$ maps from an example $x$ and any subset of the training dataset $S' \subset S$ to the correct-class margin on $x$ after training (using any fixed learning algorithm) on $S'$. Now, we can define the *sensitivity* of an example $x$ to a feature $\phi$ as:

**Definition 3** (Sensitivity to a feature $\phi$)**.** *Given a feature $\phi : \mathcal{X} \to \{0, 1\}$ with support $\phi(S) \subset S$, the* underline{sensitivity} *of an input $x$ to the feature $\phi$ is*

$$s_\phi(x, k) := \mathbb{E}_{S' \sim \alpha S}\left[ f(x; S') \,\middle|\, |\phi(S')| = k + 1 \right] - \mathbb{E}_{S' \sim \alpha S}\left[ f(x; S') \,\middle|\, |\phi(S')| = k \right], \quad (1)$$

*where for any $\alpha \in (0, 1)$, we let $S' \sim \alpha S$ represent sampling a random $\alpha$-fraction subset of $S$.*

Intuitively, $s_\phi(x, k)$ captures the effect on example $x$ of adding one more example with feature $\phi$ to the training set. Note that while $s_\phi(x, k)$ is technically a function of $\alpha$, we drop this dependence in our notation since we do not vary $\alpha$ throughout our analysis—while all our results hold for any $\alpha \in (0, 1)$, in our experiments we simply fix $\alpha = \frac{1}{2}$.

To provide further intuition for Definition 3, in Figure 2 we illustrate examples' sensitivity to a backdoor feature. In particular, for two datasets containing a backdoor trigger, we randomly select a few inputs both containing and not containing the backdoor trigger. We plot models' margin on these examples as a function of the number of backdoor examples included in the training set. For any $k \in \mathbb{N}$ and $x \in S$, the sensitivity $s_{\phi_p}(x, k)$ as defined in (1) is simply the slope at $x$-value $k$ of the corresponding line in Figure 2.

We are now ready to give a formal version of Informal Assumption 1:

**Assumption 1.** *Let $\phi_p$ be the backdoor feature, and let $S_p$ be the support (i.e., the training examples containing a backdoor). Then, for some $\alpha \in (0, 1)$[2] and all features $\phi$ with $p := |\phi_p(S)| = |\phi(S)|$,*

$$\sum_{x \in \phi_p(S)} s_{\phi_p}(x, \alpha \cdot p) \geq \delta + \sum_{x \in \phi(S)} s_\phi(x, \alpha \cdot p).$$

This assumption directly inspires our algorithm for detecting backdoor attacks.

### 3.1 APPROXIMATING SENSITIVITY USING DATAMODELS

Our goal now is to translate Assumption 1 into an algorithm for detecting backdoor training examples. To accomplish this goal, we need a way of estimating model sensitivity $s_\phi(x, k)$ for a given feature

---

[2]Again, throughout our experiments we simply fix $\alpha = \frac{1}{2}$.

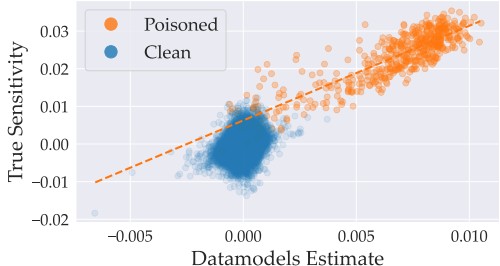 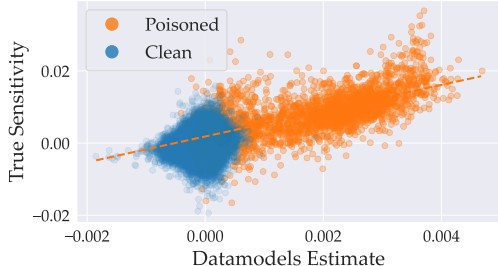

**Figure 3:** We plot for two different experiments the sensitivities ($y$ axis) and the corresponding datamodel approximations ($x$ axis), computed with (4). We observe—particularly for poisoned samples (in orange)—a good linear correlation between sensitivities and the corresponding approximations.

$\phi$ and example $x$. Now, if we had a specific feature $\phi$ in mind, we could simply compute (1) (from Definition 3) directly. In our case, however, finding the feature $\phi$ of interest is precisely our goal. To this end, in this section we show how to estimate sensitivity for all possible features $\phi$ *simultaneously*.

Our key tool here will be the recently proposed *datamodeling* framework (Ilyas et al., 2022). In particular, Ilyas et al. (2022) show experimentally that (when $f$ corresponds to training a deep neural network) for every example $x$, there exists a weight vector $w_x \in \mathbb{R}^{|S|}$ such that for subsets $S' \sim \alpha S$,

$$\mathbb{E}[f(x; S')] \approx \mathbf{1}_{S'}^\top w_x, \tag{2}$$

where $\mathbf{1}_{S'} \in \{0, 1\}^{|S|}$ is the *indicator vector* of $S'$. In other words, we can approximate the outcome of training a deep neural network on a given subset $S' \subset S$ as a linear function of the presence of each training data example.

It turns out that this property alone—captured as a formal assumption below—suffices to estimate the sensitivity of any example $x$ to any feature $\phi$.

**Assumption 2** (Datamodel accuracy). *For any example $x$, let $w_x$ be the corresponding datamodel weight. We assume that the datamodel approximates model outputs well on unseen sets, i.e., that*

$$\mathbb{E}_{S' \sim \alpha S} \left[ \left( \mathbb{E}[f(x; S')] - \mathbf{1}_{S'}^\top w_x \right)^2 \right] \leq \varepsilon. \tag{3}$$

Assumption 2 essentially guarantees that datamodels provide an accurate estimate of the average margin for random subsets $S' \sim \alpha S$. Note that we can in fact verify this assumption by sampling sets $S'$ and computing (3) directly (replacing the inner expectation with an empirical average). It turns out that this Assumption alone suffices to estimate the senstivity of an example $x$ to a feature $\phi$, as shown in the following Lemma.

**Lemma 1.** *For a feature $\phi$, let $\mathbf{1}_{\phi(S)}$ be the indicator vector of its support, and let $h : \mathbb{R}^n \to \mathbb{R}^n$ be*

$$h(v) = \frac{1}{\|v\|_1} v - \frac{1}{n - \|v\|_1} (\mathbf{1}_n - v).$$

*Then, given that Assumption 2 holds, we have that for all $x \in S$ there exists some $C > 0$ such that*

$$\left| s_\phi(x, \alpha \cdot |\phi(S)|) - w_x^\top h(\mathbf{1}_{\phi(S)}) \right| \leq C \varepsilon^{1/2} n^{1/4}. \tag{4}$$

*Proof.* See Appendix A. □

Lemma 1 provides a closed-form expression—involving only the datamodel $w_x$—for the (approximate) sensitivity of an example $x$ to a feature $\phi$.

We validate our assumptions in Figure 3 by comparing approximate sensitivities (derived from Lemma 1) to true example sensitivities (i.e., the slopes in Figure 2). Our results show that sensitivities as estimated by datamodels (as in (4)) are good approximations of ground-truth sensistivities. (We refer the reader to Appendix D for details on how we calculate ground-truth sensistivities.)

## 3.2 Poisoned Samples as a maximum-sum submatrix problem

In the previous section, we showed how to leverage datamodels (and in particular, Assumption 2) to estimate model sensitivity. In particular, Lemma 1 tells us that (assuming they are accurate) we can use datamodels to approximate examples' sensitivity to all possible features $\phi$ simultaneously. In this section, we transform Lemma 1—in combination with Assumption 1 (i.e., that the backdoor constitutes the strongest feature in the dataset)—into an algorithm that provably finds backdoor training examples.

To this end, recall that $n = |S|$ and $p = |\phi_p(S)|$. Assumption 1 says that $\sum_{x \in \phi_p(S)} s_{\phi_p}(x, \alpha \cdot p)$ is large—from here, we would like to obtain an optimization procedure that allows us to identify poisoned examples. Inspired by Lemma 1, we consider the following optimization problem:

$$\arg \max_{v \in \{0,1\}^n} h(v)^\top \mathbf{W} v \qquad \text{s.t.} \qquad \|v_i\|_1 = p, \tag{5}$$

where $\mathbf{1}_n$ is the $n$-dimensional all-ones vector and $h$ is as defined in Lemma 1. Indeed, we show in the following Lemma that under the "strongest feature" assumption (Assumption 1), the solution to (5) is precisely the indicator vector of the backdoor feature:

**Lemma 2.** *Suppose Assumption 1 holds with $\delta = \delta^*$ and Assumption 2 holds with $C = C^*$. Then if $\delta^* > 2C^* \varepsilon^{1/2} n^{1/4}$, the unique maximizer of (5) is the vector $v_p = \mathbf{1}_{\phi_p(S)}$, i.e. the indicator of the backdoor train examples.*

*Proof.* See Appendix B. $\qquad\qquad\qquad\qquad\qquad\qquad\qquad\qquad\qquad\qquad\qquad\qquad\qquad\square$

The fact that for $v \in \{0,1\}^n$ we have $\mathbf{1}_n^\top \mathbf{W} v = v^\top (\text{diag}(\mathbf{1}_n^\top \mathbf{W})) v$ allows us to express (5) as a submatrix-sum maximization problem. In particular, we have that

$$\begin{aligned}
&\underset{v \in \{0,1\}^n : \|v\|_1 = p}{\text{argmax}} \left( \frac{1}{p} \cdot v - \frac{1}{n-p} \cdot (\mathbf{1}_n - v) \right)^\top \mathbf{W} v \\
=\ &\underset{v \in \{0,1\}^n : \|v\|_1 = p}{\text{argmax}} \left( v^\top \mathbf{W} - \frac{p}{n} \cdot \mathbf{1}_n^\top \mathbf{W} \right) v \\
=\ &\underset{v \in \{0,1\}^n : \|v\|_1 = p}{\text{argmax}} \ v^\top \left( \mathbf{W} - \text{diag}\left( \frac{p}{n} \cdot \mathbf{1}_n^\top \mathbf{W} \right) \right) v.
\end{aligned} \tag{6}$$

In the next section, we demonstrate how to efficiently approximate a solution to (6), and propose a heuristic to detect backdoor samples that does not require knowledge of the size $p$.

## 4 Detecting poisoned samples with datamodels

In the previous section, we reduced the problem of finding the backdoor training examples to that of finding a submatrix of maximal sum within the datamodel matrix. However, the formulation presented in (5) is difficult to solve for multiple reasons. First, the optimization problem requires knowledge of the number of poisoned samples $|\phi_p(S)|$, which in practice is unknown. Second, even if we did know number of poisoned samples, the problem is still NP-hard in general (Branders et al., 2017). In fact, even linearizing (5) and using the commercial-grade mixed-integer linear program solver Gurobi (Gurobi Optimization, LLC, 2021) takes several days to solve (per problem instance) due to the sheer number of variables being optimized over.

We thus resort to approximation. For each $k$ in a pre-defined set of "candidate sizes" $K$ for the submatrix in (6), we set $p$ equal to $k$. We then solve the resulting maximization problem with a greedy local search algorithm inspired by the Kernighan-Lin heuristic for graph partitioning (Kernighan & Lin, 1970). That is, starting from a random assignment for $v \in \{0,1\}^n$, the algorithm considers all pairs of indices $i, j \in [n]$ such that $v_i \neq v_j$, and such that swapping the values of $v_i$ and $v_j$ would improve the submatrix sum objective. The algorithm greedily selects the pair that would most improve the objective and terminates when no such pair exists. We run this local search algorithm $T = 1000$ times for each value of $k$ and collect the *candidate solutions* $\{\boldsymbol{v}^{k,l} : k \in K, l \in [T]\}$.

Now, rather than using any particular one of these solutions, we combine them to yield a score for each example $x_i \in S$. We define this score for the $i$-th example $x_i$ as the weighted sum of the number of times it was included in the solution of local search, that is,

$$s_i = \sum_{k \in K} \frac{1}{k} \sum_{l=1}^{T} \boldsymbol{v}_i^{k,l}. \tag{7}$$

Intuitively, we expect that backdoor training examples will end up in many of the greedy local search solutions and thus have a high score $s_i$. We translate the scores (7) into a concrete algorithm by flagging (and removing) the examples with the highest score.

## 5 EXPERIMENTS AND RESULTS

In the previous sections, we presented a theoretical framework to detect backdoor examples in a dataset. In this section, we start by validating our theoretical framework against two common types of backdoor attacks: dirty-label attacks (Gu et al., 2017) and clean-label attacks (Turner et al., 2019). For each poisoning setup, we validate how well the theory aligns with what we actually observe. We then apply our defense from Section 4 and measure its effectiveness in defending against the mounted attacks. Table 1 shows a summary of our experiments, and more details are provided in Appendix D.

We compare our algorithm against multiple baselines: Inverse Self-Paced Learning[3] (ISPL) (Jin et al., 2021), Spectral Signatures[4] (SS) (Tran et al., 2018), SPECTRE[5] (Hayase et al., 2021a) and Activation Clustering[6] (AC) (Chen et al., 2018). To evaluate the algorithms, we use the CIFAR-10 dataset (Krizhevsky, 2009) and mount the attacks presented in Table 1.

**Compute Cost.** For all of our experiments, we use the ResNet-9 model (He et al., 2015)[7]. In order to compute the datamodels, we employ the framework presented in (Ilyas et al., 2022). Specifically, for each one of our experiments, we train 100,000 models on CIFAR10. Each model is trained on 50% of the dataset (chosen uniformly at random), and requires $\sim 40$ seconds for convergence. In total, training all the models for one experiment requires roughly 1 day using 16 V100 GPUs.

**Table 1:** To evaluate the effectiveness of our framework, we run multiple experiments and vary the attack type, the trigger, the target class, and the proportion of poisoned samples. We report the clean accuracy and the poisoned accuracy of every attack when no defense is employed. A larger gap between clean and poisoned accuracy means the attack is more effective. A sample of the triggers used is presented in Figure 5.

| Exp. | Attack Type | Frac. | Clean Acc. | Poisoned Acc. |
|------|------------|-------|-----------|---------------|
| 1 | Dirty Label | 1.5% | 86.64 | 19.90 |
| 2 | Dirty-Label | 5% | 86.67 | 12.92 |
| 3 | Dirty-Label | 1.5% | 86.39 | 49.57 |
| 4 | Dirty-Label | 5% | 86.23 | 10.67 |
| 5 | Clean-Label | 1.5% | 86.89 | 75.58 |
| 6 | Clean-Label | 5% | 87.11 | 41.89 |
| 7 | Clean-Label (no adv.) | 5% | 86.94 | 71.68 |
| 8 | Clean-Label (no adv.) | 10% | 87.02 | 52.08 |

---

[3]We thank the authors for sharing their implementation with us.
[4]We re-implement the algorithm presented in the paper.
[5]https://github.com/SewoongLab/spectre-defense
[6]We use the implementation provided in: https://github.com/SewoongLab/spectre-defense
[7]https://github.com/wbaek/torchskeleton/blob/master/bin/dawnbench/cifar10.py

## 5.1 VERIFYING OUR ASSUMPTIONS

Our theory in Section 3 indicates that we can approximate the sensitivity $s_\phi(x, \cdot)$ of an example $x$ to the backdoor feature using datamodels. Furthermore, Assumption 1 tells us that the model is more sensitive towards the backdoor feature than towards other features.

To verify these assumptions, we compute the vector $h(\mathbf{1}_{\phi_p(S)})$ presented in Lemma 1, and multiply it by the datamodels matrix $\mathbf{W}$. We then check how much this product correlates with the ground truth poison mask by computing the area under the ROC curve (AUROC) between these two quantities. Note that this does not constitute a valid defense, since it requires knowledge of the ground truth. However, the purpose of this experiment is to show the best results we could hope to achieve by employing our method.

As we can see in Table 2, we observe a high AUROC in seven out of eight poisoning setups, which validates our assumption that the poison feature is among the strongest features. Furthermore, we observe that the AUROC for Exp. 2 is low, and hence our assumptions do not hold in that setup. Indeed, in Figure 6 (Appendix E) we see that the for Exp. 2, the sensitivity $s_{\phi_p}(x, \alpha \cdot |\phi_p(S)|) \approx 0$ even for $x \in \phi_p(S)$. We hypothesize that the problem is the large number of poisoned samples in that experiment and as a result, adding more poisoned samples has little effect on the predictions of a model for backdoor examples. One potential solution is to train the models on smaller subsets of the dataset when estimating datamodels, e.g. 20% of the dataset.

**Table 2:** We measure the AUROC between the sensitivity metric from (4) and the ground truth poison mask. We observe a large AUROC in seven out of eight experiments, which indicates that sensitivity measured using datamodels indeed correlates with the choice of poisoned samples.

| Exp. 1 | Exp. 2 | Exp. 3 | Exp. 4 | Exp. 5 | Exp. 6 | Exp. 7 | Exp. 8 |
|--------|--------|--------|--------|--------|--------|--------|--------|
| 99.99  | 60.88  | 97.98  | 97.71  | 99.95  | 99.98  | 96.98  | 98.26  |

## 5.2 EVALUATING THE EFFECTIVENESS OF OUR PROPOSED DEFENSE

To evaluate the effectiveness of our proposed defense, we run our algorithm in the eight different poisoning setups described in Table 1. After computing the scores returned by our defense algorithm, we compare these scores with the ground truth mask, and measure their corresponding AUROC.

**Table 3:** We measure the AUROC between our algorithm's scores and ground truth.

| Exp. 1 | Exp. 2 | Exp. 3 | Exp. 4 | Exp. 5 | Exp. 6 | Exp. 7 | Exp. 8 |
|--------|--------|--------|--------|--------|--------|--------|--------|
| 94.25  | 67.44  | 74.35  | 80.18  | 93.37  | 93.23  | 91.14  | 95.51  |

As we can see, in Table 3, our method's scores are highly predictive of the poisoned samples in seven out of eight poisoning setups (in fact, the setup that fails is exactly the one predicted by Section 5.1).

We remove from the training dataset 10% the samples with the highest scores, train a model on the curated dataset, and test how well it performs on a clean and a poisoned validation sets. We report our results in Table 4. As we can see, our defense is successful in defending against seven out of eight poisoning setups.

## 6 RELATED WORK

This work focuses on backdoor data poisoning attacks. First, we briefly contrast these attacks with several other popular attacks on ML models.

*Targeted data poisoning attacks* are a closely related, but different, threat model, where the goal of the adversary is to misclassify a pre-defined test sample, by modifying only the training dataset (Koh & Liang, 2017; Shafahi et al., 2018). In contrast, backdoor attacks can be applied to any image at inference time by inserting the backdoor trigger.

**Table 4:** We compare our method against multiple baselines in a wide range of experiments. We observe that our algorithm achieves a high accuracy on the fully poisoned validation set in seven out of eight experiments. Refer to Table 1 for the full experiments parameters.

| Exp. | No Defense | | AC | | ISPL | | SPECTRE | | SS | | Ours | |
|---|---|---|---|---|---|---|---|---|---|---|---|---|
| | Clean | Poisoned | Clean | Poisoned | Clean | Poisoned | Clean | Poisoned | Clean | Poisoned | Clean | Poisoned |
| 1 | 86.64 | 19.90 | 86.76 | 19.68 | 86.13 | **86.15** | 86.71 | 20.17 | 85.52 | 30.99 | 85.05 | **85.06** |
| 2 | 86.67 | 12.92 | 85.41 | 12.93 | 85.88 | **85.82** | - | - | 85.33 | 13.63 | 84.61 | 25.24 |
| 3 | 86.39 | 49.57 | 86.25 | 48.85 | 86.32 | **85.57** | 86.28 | 45.32 | 85.22 | 78.22 | 84.82 | **84.11** |
| 4 | 86.23 | 10.67 | 84.75 | 10.82 | 85.86 | **85.18** | - | - | 84.85 | 13.33 | 84.64 | **83.72** |
| 5 | 86.89 | 75.58 | 86.73 | 82.83 | 86.04 | **85.89** | 86.82 | 80.65 | 85.67 | **85.41** | 83.82 | 83.72 |
| 6 | 87.11 | 41.89 | 86.85 | 51.05 | 86.18 | **86.11** | 86.97 | 51.18 | 85.68 | **85.60** | 84.88 | 84.79 |
| 7 | 87.02 | 71.68 | 86.90 | 73.28 | 86.50 | 82.31 | 86.72 | 76.97 | 85.70 | 82.70 | 84.19 | **84.02** |
| 8 | 86.94 | 52.08 | 86.81 | 56.78 | 86.04 | 71.27 | 86.63 | 52.27 | 85.87 | 71.93 | 84.81 | **84.66** |

*Indiscrimate (or availability) attacks* (Muñoz-González et al., 2017; Lu et al., 2022) are another popular threat model. During an indiscrimante attack, the goal of the attacker is to degrade the test performance as much as possible by poisoning a small fraction of the training set. In contrast, backdoor attacks are designed to only affect test images once the trigger is inserted.

Developing backdoor attacks and defenses in the context of deep learning is an increasingly active area of research (Gu et al., 2017; Tran et al., 2018; Chen et al., 2018; Turner et al., 2019; Saha et al., 2020; Shokri et al., 2020; Hayase et al., 2021b; Qi et al., 2022; Goldblum et al., 2022; Goldwasser et al., 2022) (see e.g. (Li et al., 2022) for a survey).

A popular line of work for defending against backdoor attacks applies outlier detection in the latent space of neural networks (Tran et al., 2018; Chen et al., 2018; Hayase et al., 2021b). Such defenses inherently fail in defending against adaptive attacks that completely bypass the defense while maintaining the attack efficacy (Shokri et al., 2020).

While we assume that backdoors are the strongest features, other defenses make explicit assumptions about the triggers themselves. For example, Wang et al. (2019) assume that triggers are small in norm. Some of the attacks we consider violate the small-norm assumption, as shown in Figure 1a.

A recent line of work investigates certified defenses against backdoor attacks (Levine & Feizi, 2021; Wang et al., 2022). The authors provide certificates by training separate models on different partitions of the training set, and dropping the models trained on data containing poison. This approach, however, significantly degrades the accuracy of the trained model, and provides certificates against a prohibitively small number of poisoned samples.

A number of prior works explore the applicability of influence-based methods as defenses against different attacks in deep learning (Koh & Liang, 2017). To the best of our knowledge, only Lin et al. (2022) discuss using such methods for backdoor attacks. However, their defense requires knowledge of the (typically unknown) attack parameters. Closest to our work is that of Jin et al. (2021), who (like this work) consider a defense based on model behavior rather than properties of any latent space.

## 7 CONCLUSION

In this paper, we proposed a new perspective of data poisoning. Specifically, we argued that backdoor attacks are fundamentally indistinguishable from existing features in the data. Consequently, the task of detecting backdoor examples in a dataset is equivalent to that of detecting features that models are particularly sensitive to. Based on this observation, we propose a framework—and corresponding algorithm—for identifying backdoor examples in training data. Through a wide range of backdoor poisoning experiments, we demonstrated the effectiveness of our approach in defending against backdoor attacks, while retaining high accuracy.

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

## A   PROOF OF LEMMA 1

We separate the proof in two parts. First, we show that $s_\phi$ can be accurately approximated with datamodels. In particular,

**Lemma 3.** *Suppose $\alpha$ is such that $c \leq \alpha \leq 1 - c$ for some constant $c \in (0,1)$. Then, there exists a constant $C > 0$ such that for all $x \in S$ we have*

$$\left| s_\phi(x, \alpha p) - \mathbb{E}_{S' \sim \alpha S} \left[ w_x^\top \mathbf{1}_{S'} \,\middle|\, |\phi(S')| = \alpha p \right] + \mathbb{E}_{S' \sim \alpha S} \left[ w_x^\top \mathbf{1}_{S'} \,\middle|\, |\phi(S')| = \alpha p + 1 \right] \right|$$
$$\leq C\varepsilon^{1/2} n^{1/4}.$$

*Proof.* For convenience, assume that $\alpha p$ is an integer. Using the triangle inequality, it is enough to show that

$$\left| \mathbb{E}_{S' \sim \alpha S} \left[ f(x; S') \,\middle|\, |\phi(S')| = \alpha p \right] - \mathbb{E}_{S' \sim \alpha S} \left[ w_x^\top \mathbf{1}_{S'} \,\middle|\, |\phi(S')| = \alpha p \right] \right| \qquad (8)$$
$$\leq C\varepsilon^{1/2} n^{1/4}/2$$

and

$$\left| \mathbb{E}_{S' \sim \alpha S} \left[ f(x; S') \,\middle|\, |\phi(S')| = \alpha p + 1 \right] - \mathbb{E}_{S' \sim \alpha S} \left[ w_x^\top \mathbf{1}_{S'} \,\middle|\, |\phi(S')| = \alpha p + 1 \right] \right| \qquad (9)$$
$$\leq C\varepsilon^{1/2} n^{1/4}/2.$$

Starting with Equation (8), we have

$$\mathbb{P}_{S' \sim \alpha S}[|\phi(S')| = \alpha p] = \frac{\binom{p}{\alpha p} \binom{n-p}{\alpha(n-p)}}{\binom{n}{\alpha n}}$$

and

$$\mathbb{P}_{S' \sim \alpha S}[|\phi(S')| = \alpha p + 1] = \frac{\binom{p}{\alpha p + 1} \binom{n-p}{\alpha(n-p) - 1}}{\binom{n}{\alpha n}}$$
$$= \left( \frac{p(1-\alpha)}{\alpha p + 1} \cdot \frac{\alpha(n-p)}{(1-\alpha)(n-p) + 1} \right) \cdot \mathbb{P}_{S' \sim \alpha S}[|\phi(S')| = \alpha p]$$

We first show that the ratio of the two probabilities is bounded by a constant, i.e.

$$\frac{p(1-\alpha)}{\alpha p + 1} \cdot \frac{\alpha(n-p)}{(1-\alpha)(n-p) + 1}$$
$$= \frac{\alpha}{\alpha + \frac{1}{p}} \cdot \frac{1-\alpha}{1 - \alpha + \frac{1}{n-p}}$$
$$\geq \frac{\alpha}{\alpha + \alpha} \cdot \frac{1-\alpha}{2 - \alpha} \geq \frac{1}{2} \cdot \frac{c}{1+c},$$

where we used that $1 \leq \alpha p$ and $\alpha \leq 1 - c$. Thus

$$\mathbb{P}_{S' \sim \alpha S}[|\phi(S')| = \alpha p + 1] \geq \frac{c}{2(1+c)} \mathbb{P}_{S' \sim \alpha S}[|\phi(S')| = \alpha p].$$

Now we proceed with boudning $\mathbb{P}_{S' \sim \alpha S}[|\phi(S')| = \alpha p]$. Using Stirling's approximation, we have

$$\mathbb{P}_{S' \sim \alpha S}[|\phi(S')| = \alpha p] \asymp \sqrt{\frac{n}{p(n-p)} \frac{1}{\alpha(1-\alpha)}} \geq \frac{C^2}{\sqrt{n}}$$

for some constant $C$. Now from the triangle inequality, Jensen's inequality and Markov's inequality we have that for sufficiently large $n$

$$\mathbb{E}_{S' \sim \alpha S} \left[ f(x; S') \,\middle|\, |\phi(S')| = \alpha p \right] - \mathbb{E}_{S' \sim \alpha S} \left[ w_x^\top \mathbf{1}_{S'} \,\middle|\, |\phi(S')| = \alpha p \right]$$

$$\leq \mathbb{E}_{S' \sim \alpha S} \left[ \left| f(x; S') - w_x^\top \mathbf{1}_{S'} \right| \,\middle|\, |\phi(S')| = \alpha p \right]$$

$$\leq \sqrt{\mathbb{E}_{S' \sim \alpha S} \left[ (f(x; S') - w_x^\top \mathbf{1}_{S'})^2 \,\middle|\, |\phi(S')| = \alpha p \right]}$$

$$\leq C \varepsilon^{1/2} n^{1/4}.$$

The case for Equation (9) is analagous. $\qquad \square$

Next, we show that $w_x^\top h(\mathbf{1}_{\phi(S)})$ corresponds to the desired difference of conditional expectations. In this proof, we let $h_\phi = h(\mathbf{1}_{\phi(S)})$ for brevity.

**Lemma 4.** *We have for every $x \in S$ that*

$$\mathbb{E}_{S' \sim \alpha S} \left[ w_x^\top \mathbf{1}_{S'} \,\middle|\, |\phi(S')| = \alpha p \right] - \mathbb{E}_{S' \sim \alpha S} \left[ w_x^\top \mathbf{1}_{S'} \,\middle|\, |\phi(S')| = \alpha p + 1 \right] = w_x^\top h_\phi.$$

*Proof.* Note that we can write

$$\mathbb{E}_{S' \sim \alpha S} \left[ w_x^\top \mathbf{1}_{S'} \,\middle|\, |\phi(S')| = \alpha p \right]$$

$$= \mathbb{E}_{S' \sim \alpha S} \left[ \sum_{z \in S} \mathbf{1}_{z \in S} w_{xz} \,\middle|\, |\phi(S')| = \alpha p \right]$$

$$= \sum_{z \in S'} \mathbb{P}_{S' \sim \alpha S} \left[ z \in S \,\middle|\, |\phi(S')| = \alpha p \right] \cdot w_{xz}.$$

There are a total of

$$\binom{p}{\alpha p} \binom{n - p}{\alpha(n - p)}$$

sets satisfying $|\phi(S')| = \alpha p$, and each is sampled with the same probability. Among these, given $z \in S'$ there are

$$\binom{p - 1}{\alpha p - 1} \binom{n - p}{\alpha(n - p)}$$

sets containing $z$, so for all $z \in S'$ we have

$$\mathbb{P}\left[ z \in S' \,\middle|\, |\phi(S')| = \alpha p \right] = \frac{\alpha p}{p}.$$

Similarly, we have for all $z$ in the complement $(S')^c$ that

$$\mathbb{P}\left[ z \in S' \,\middle|\, |\phi(S')| = \alpha p \right] = \frac{\alpha(n - p)}{n - p}.$$

Thus, overall

$$\mathbb{E}_{S' \sim \alpha S} \left[ w_x^\top \mathbf{1}_{S'} \,\middle|\, |\phi(S')| = \alpha p \right] = \frac{\alpha p}{p} w_x^\top \mathbf{1}_{S'} + \frac{\alpha(n - p)}{n - p} w_x^\top \mathbf{1}_{(S')^c}$$

Analogously,

$$\mathbb{E}_{S' \sim \alpha S} \left[ w_x^\top \mathbf{1}_{S'} \,\middle|\, |\phi(S')| = \alpha p + 1 \right] = \frac{\alpha p + 1}{p} w_x^\top \mathbf{1}_{S'} + \frac{\alpha(n - p) - 1}{n - p} w_x^\top \mathbf{1}_{(S')^c}$$

and the lemma follows when we subtract the two. $\qquad \square$

The proof of Lemma 1 follows by combining the results of Lemma 3 and Lemma 4.

# B  PROOF OF LEMMA 2

*Proof.* The result follows directly from Assumption 1 and Assumption 2. In particular, let $\phi_v$ be a feature whose corresponding support is $\phi_v(S) = \text{supp}(v)$.

$$\left(\frac{1}{n} \cdot v - \frac{1}{n-p} \cdot (\mathbf{1}_n - v)\right)^\top \mathbf{W}v = \begin{bmatrix} h_v^\top w_{x_1} \\ h_v^\top w_{x_2} \\ \cdots \\ h_v^\top w_{x_n} \end{bmatrix}^\top v = \sum_{i \in \phi_v(S)} h_v^\top w_{x_i}. \tag{10}$$

First, from Assumption 2 we have that $\sum_{i \in \phi_v(S)} h_v^\top w_{x_i} \geq \sum_{i \in \phi_v(S)} s_v(x_i, \alpha \|v\|_1) - pC\varepsilon^{1/2} n^{1/4}$.

Now let $v_p$ be such that $\text{supp}(v_p) = \phi_p(S)$, i.e. $v_p$ is the indicator for the poisoned examples. From Assumption 1 we have that $v_p \sum_{i \in \phi_v(S)} s_v(x_i, \alpha \|v\|_1) - pC\varepsilon^{1/2} n^{1/4}$.

Analougously, we have that $\sum_{i \in \phi_v(S)} h_v^\top w_{x_i} \leq \sum_{i \in \phi_v(S)} s_v(x_i, \alpha \|v\|_1) + pC\varepsilon^{1/2} n^{1/4}$. Thus for any $v \neq v_p$ we have that

$$\left(\frac{1}{n} \cdot v_p - \frac{1}{n-p} \cdot (\mathbf{1}_n - v_p)\right)^\top \mathbf{W}v_p - \left(\frac{1}{n} \cdot v - \frac{1}{n-p} \cdot (\mathbf{1}_n - v)\right)^\top \mathbf{W}v \tag{11}$$

$$\geq \sum_{i \in \phi_{v_p}(S)} s_{v_p}(x_i, \alpha \|v_p\|_1) - \sum_{i \in \phi_v(S)} s_v(x_i, \alpha \|v\|_1) + 2pC\varepsilon^{1/2} n^{1/4} \tag{12}$$

$$\geq p\delta - 2pC\varepsilon^{1/2} n^{1/4} > 0, \tag{13}$$

where the last inequality follows by the assumption that $\delta > 2C\varepsilon^{1/2} n^{1/4}$. $\qquad\square$

## C  Naturally-occuring triggers

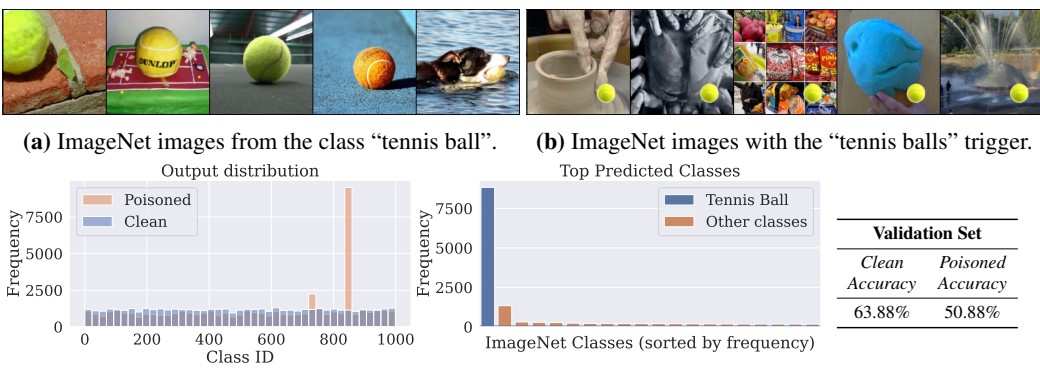

**(a)** ImageNet images from the class "tennis ball".  **(b)** ImageNet images with the "tennis balls" trigger.

| | Validation Set | |
|---|---|---|
| | *Clean Accuracy* | *Poisoned Accuracy* |
| | 63.88% | 50.88% |

**(d)** Model accuracy

**(c)** Predictions of a ResNet-18 on the fully poisoned validation set

**Figure 4:** An adversary can even leverage the features of a dataset to mount a backdoor attack. (a) The tennis ball feature is already present in the ImageNet training set, so we do *not* modify the dataset at all. (b) Instead, we show that a small picture of a tennis ball is a "pre-existing" backdoor trigger. Plots (c) and (d) are the same as Figures (1b) and (1c).

We have shown in Section 2 that triggers and features are inherently indistinguishable. Specifically, the adversary could use an arbitrary feature in order to mount a backdoor attack (c.f. Figure 4).

In fact, the adversary could also leverage existing (naturally occurring) features to mount backdoor attacks. In particular, the adversary can choose a feature that has a high correlation with a particular class, and leverage this bias to mount a backdoor attack. For example, we show in Figure 4 that we can use tennis balls as triggers for models pretrained on ImageNet. By pasting tennis balls on the validation set of ImageNet, the accuracy of the pretrained model drops from 63.88% to 50.88%. Furthermore, the most predicted class for the poisoned dataset is "Tennis Ball".

# D EXPERIMENTAL SETUP

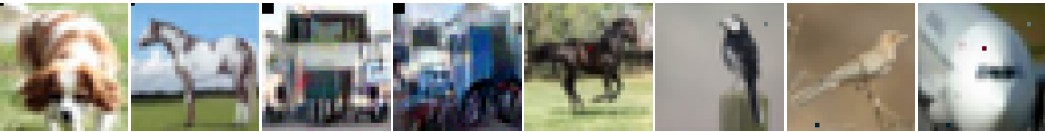

**Figure 5:** We execute the poisoning attacks with three types of triggers: (a) one black pixel on top left corner (first two images), (b) 3x3 black square on top left corner (third and fourth images), and (c) 3-way triggers adapted from (Xie et al., 2020) (last four images).

## D.1 BACKDOOR ATTACKS

**Dirty-Label Backdoor Attacks.** The most prominent type of backdoor attacks is a dirty-label attack (Gu et al., 2017). During a dirty-label attack, the adversary inserts a trigger into a subset of the training set, then flips the label of the poisoned samples to a particular target class $y_b$. We mount four different dirty-label attacks, by considering two different triggers, and two different levels of poisoning (c.f. Exp. 1 to 4 in Table 1).

**Clean-Label Backdoor Attacks.** A more challenging attack is the clean-label attack (Shafahi et al., 2018; Turner et al., 2019)[8] where the adversary avoids changing the label of the poisoned samples. To mount a successful clean-label attack, the adversary poisons samples from the target class only, hoping to create a strong correlation between the target class and the trigger.

We perform two types of clean-label attacks. During the first type (Exp. 5 and 6 from Table 1), we perturb the image with an adversarial example before inserting the trigger, as presented in (Turner et al., 2019). During the second type of clean-label attacks (Exp. 7 and 8 from Table 1), we avoid adding the adversarial example, however, we poison more samples to have an effective attack.

**Trigger.** We conduct our experiments with two types of triggers. The first type is a fixed pattern inserted in the top left corner of the image. The trigger is unchanged between train and test time. This type of trigger has been used in multiple works (Gu et al., 2017; Turner et al., 2019). The other type of trigger is an m-way trigger, with m=3 (Xie et al., 2020). During training, one of three triggers is chosen at random for each image to be poisoned, and then the trigger is inserted into one of three locations in the image. At test time, all three triggers are inserted at the corresponding positions to reinforce the signal. We display in Figure 5 the triggers used to poison the dataset.

## D.2 TRAINING SETUP

**Training CIFAR models.** In this paper, we train a large number of models on different subsets of CIFAR-10 in order to compute the datamodels. To this end, we use the ResNet-9 architecture (He et al., 2015)[9]. This smaller version of ResNets was optimized for fast training.

**Training details.** We fix the training procedure for all our models. We show the hyperparameter details in Table 5[10]. One augmentation was used for dirty-label attacks (Cutout (DeVries & Taylor, 2017)) to improve the performance of the model on CIFAR10. Similar to (Turner et al., 2019), we do not use any data augmentation when performing clean-label attacks.

**Performance.** In order to train a large number of models, we use the FFCV library for efficient data-loading (Leclerc et al., 2022). The speedup from using FFCV allows us to train a model to convergence in ∼40 seconds, and 100k models for each experiment using 16 V100 in roughly 1 day[11].

---

[8]We evaluate the clean-label attack as presented in (Turner et al., 2019)

[9]https://github.com/wbaek/torchskeleton/blob/master/bin/dawnbench/cifar10.py

[10]Our implementation and configuration files will be available in our code.

[11]We train 3 models in parallel on every V100.

**Table 5:** Hyperparameters used to train ResNet-9 on CIFAR10.

| Optimizer | Epochs | Batch Size | Peak LR | Cyclic LR Peak Epoch | Momentum | Weight Decay |
|-----------|--------|------------|---------|----------------------|----------|--------------|
| SGD | 24 | 1,024 | 0 | 5 | 0.9 | 4e-5 |

**Computing datamodels.** We adopt the framework presented in (Ilyas et al., 2022) to compute the datamodels of each experiment. Specifically, we train 100k models on different subsets containing 50% of the training set chosen at random. We then compute the datamodels as described in (Ilyas et al., 2022).

**Local Search.** We approximate the solution of the problem outlined in (5) using a local search heuristic presented in (Kernighan & Lin, 1970). We iterate over ten sizes for the poison mask: {10, 20, 40, 80, 160, 320, 640, 1280, 2560, 5120}. For each size, we collect 1,000 different solutions by starting from different initialization of the solution.

### D.3 ESTIMATING THEORETICAL QUANTITIES

Recall the average margin definition presented in (1). In particular:

$$\mathbb{E}_{S' \sim \alpha S} \left[ f(x; S') \, \middle| \, |\phi(S')| = k \right] \tag{14}$$

where $S'$ is a subset of the training set, $f(x; S')$ is the margin of the model on $x$ when trained on the dataset $S'$, $\phi(S')$ is the subset of the set $S'$ containing the poisoned feature, and $k$ is the number of poisoned samples. Estimating the average margins requires training a large number of models on different subsets, and measure–for every sample $x$ and every number of poisoned samples $k$–the margins of the trained model.

For the purpose of this paper, we leverage the datamodels computation framework to estimate these averages. In particular, to compute the datamodels weights, we train a large number of models on different subsets $S_1, S_2, \ldots, S_T$ of the training set[12]. For every subset $S_i$, we record the number of poisoned samples in the subset, then we estimate the average margin by averaging the margins over the different subsets that contain $k$ poisoned samples.

$$N_\phi(k) = \sum_{i=1}^{T} \mathbf{1}_{|\phi(S_i)|=t} \tag{15a}$$

$$\mathbb{E}_{S' \sim \alpha S} \left[ f(x; S') \, \middle| \, |\phi(S')| = k \right] \approx \frac{1}{N_\phi(k)} \sum_{i=1}^{T} f(x; S_i) \cdot \mathbf{1}_{|\phi(S_i)|=t} \tag{15b}$$

By a training 100k models on different subsets of the dataset, we obtain reasonable estimates of the marginal effects for every sample $x$.

---

[12]We refer the reader to (Ilyas et al., 2022) for more details.

# E  OMITTED PLOTS

## E.1  AVERAGE MARGIN PLOTS

In this section, we show for all the experiments the plots of the average margin for clean and poisoned samples as a function of the number of poisoned samples in the dataset (c.f. Fig. 2 in the main paper).

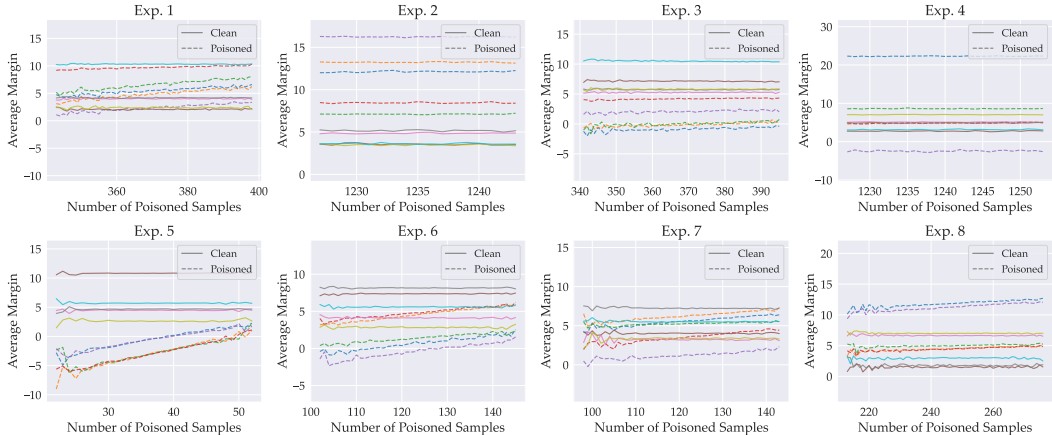

**Figure 6:** We plot for all the experiments the average margin for five clean samples (left) and five poisoned samples (right) as the number of poisoned samples in the training set increases. We observe that the average margin for *clean samples* (without the trigger) is constant when poisoning more samples in the dataset. In contrast, the average margin for *poisoned samples* (with the trigger) increases when the number of poisoned samples increases in the dataset, confirming our assumptions.

## E.2  ESTIMATED SENSITIVITIES PLOTS

In this section, we show for all the experiments the plots of the estimated sensitivities, and the approximation we obtain using datamodels (c.f. Fig. 3 in the main paper).

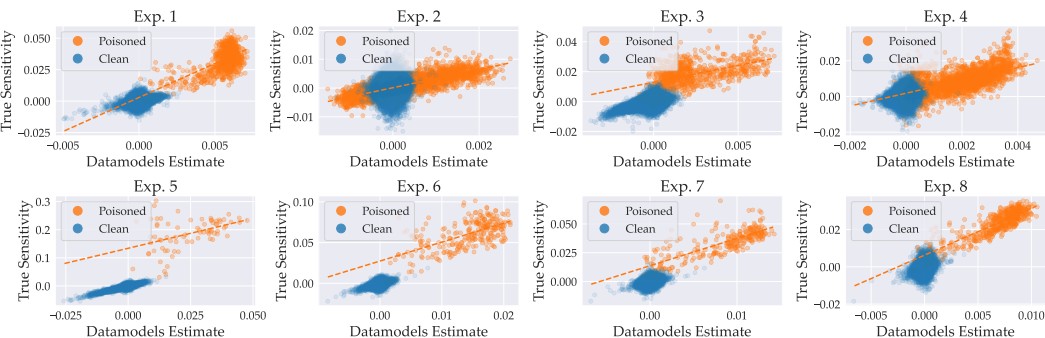

**Figure 7:** We plot for all the experiments the estimated marginal sensitivities and the approximation with datamodels presented in Equation 4. We observe for poisoned samples (in red) a good linear correlation between the sensitivities and the datamodels' approximation. Additionally, we observe no noticeable correlation for clean samples (in green).

## E.3  DISTRIBUTION OF DATAMODELS VALUES

In this section, we plot for each experiment the distribution of the datamodels weights for all experiments. In particular, recall that the datamodels weight $w_x[i]$ represents the influence of the training sample $x_i$ on the sample $x$. We show below the distribution of the effect of 1) poisoned

samples on poisoned samples, 2) the poisoned samples on the clean samples and 4) the clean samples on the clean samples.

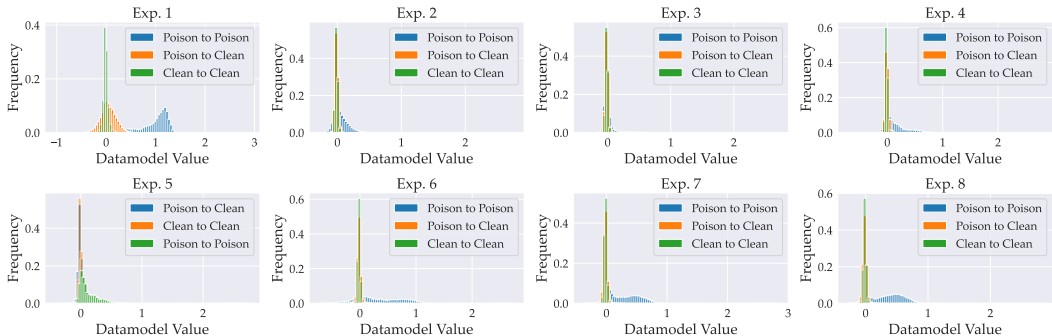

**Figure 8:** We plot the distribution of the datamodels weights for all the experiments. We clearly see that the effect of poisoned samples on other poisoned samples is generally higher than the effect of poisoned samples on clean samples, and than clean samples on each other.

### E.4   ATTACK SUCCESS RATE (ASR)

In the main paper, we presented our results by measuring the accuracy of a model on a clean and a poisoned validation sets. Another relevant metric is the Attack Success Rate (ASR) which measures the probability that the model predicts the target class after inserting the trigger into an image. As we can see in Table 6, our defense leads to a low ASR in seven out of eight setups.

**Table 6:** We compare our method against multiple baselines in a wide range of experiments. We observe that our algorithm leads to a low ASR in seven out of eight experiments. Refer to Table 1 for the full experiments parameters.

| Exp. | No Defense | AC | ISPL | SPECTRE | SS | Ours |
|------|------------|-------|----------|----------|-------|----------|
| 1 | 87.94 | 88.26 | **0.70** | 87.67 | 73.78 | **0.81** |
| 2 | 96.38 | 96.32 | **0.67** | - | 95.40 | 82.56 |
| 3 | 50.49 | 51.33 | **0.58** | 55.68 | 10.44 | **1.18** |
| 4 | 99.21 | 99.02 | **0.75** | - | 95.85 | **2.30** |
| 5 | 15.66 | 5.35 | **0.71** | 7.66 | **0.80** | 0.92 |
| 6 | 58.57 | 45.44 | **0.66** | 46.78 | **0.67** | 0.77 |
| 7 | 26.09 | 23.64 | 9.90 | 18.48 | 9.17 | **3.56** |
| 8 | 50.62 | 44.82 | 26.07 | 44.14 | 24.72 | **3.42** |

