# OpenReview forum: "Backdoor or Feature? A New Perspective on Data Poisoning"
_ICLR.cc/2023/Conference — Submitted to ICLR 2023_

### Official Review · Reviewer_9GLS · 2022-10-25

**Confidence:** 4
**Correctness:** 2
**Technical Novelty And Significance:** 2
**Empirical Novelty And Significance:** 2
**Recommendation:** 3

**Clarity, Quality, Novelty And Reproducibility:**


The perspective of defining backdoor attacks as the strongest features is
interesting.

Its definitions of features and feature supports seem to be oversimplified. In
section 3, this paper defines a feature as a map from an input to a binary
value of either 0 or 1 and defines its feature support as whether the feature
is present. These definitions overly simplify the problem and discussion. In
practice, the presence of a feature is hard to measure or define -- moreover,
a feature is also hard to define or measure. Using a single scalar value to
denote its existence or not is overly simple. Moreover, the feature itself is
an ambiguous concept. In many cases, humans cannot even determine if a feature
exists or not without enough context. For example, it mentions feature support
of a dog's feature can be whether it contains "dog fur" -- my question is, do
you consider a short white line in an image as dog fur or not? These
properties of its defined feature support also affect the decision of whether
a feature is present. This paper directly defines a feature as a
representation of having something or not, which to me, tries to convert the
analysis of a DNN on image data to the analysis of tabular data. I am not
convinced that an image in DNN can be analyzed this way. If so, some hard
problems in DNN will become easy.

Another fundamental issue of this work is that it assumes that backdoor
attacks are the strongest features without considering the learning process.
This paper assumes that backdoors as stronger features than natural features
originally occur in input samples. This is wrong. I want to point out that
many datasets do have strong natural features. In backdoor research, this is
known as the natural backdoor. The research goes back to ABS by Liu et al. CCS
2019 where the authors show natural backdoors for deers. Further, existing
work Ex-ray by Liu et al. CVPR 2022 have shown that natural backdoors, whose
feature are not as strong as input features, can still manipulate models to
perform successful backdoor attacks. Another line of work is trying to develop
a robust learning method, e.g., ABL by Li et al. NeurIPS 2021, in the face of
poisoning data. They also prove that the learning process can pick up more
desirable features in training to produce more robust models. This also shows
that the assumption of this work does not hold in practice.

One reason I think its experiment result is good is that it only considers
patch triggers which are easier to present backdoor features at the input. It
is important to consider different types of backdoor triggers like Blended
attack [1], WaNet [2], and Invisible attack [3], whose features at input space
are not obvious.

[1] Chen et al. "Targeted Backdoor Attacks on Deep Learning Systems Using Data Poisoning." arXiv 2017
[2] Nguyen et al. "WaNet - Imperceptible Warping-based Backdoor Attack" ICLR 2021
[3] Li et al. "Invisible Backdoor Attack with Sample-Specific Triggers" ICCV 2021


**Strength And Weaknesses:**

It is an interesting perspective to look at backdoors.
The paper presentation is good.

Its definition seems to be oversimplified.
The assumption it makes seems to be wrong.
Evaluation is weak.

**Summary Of The Paper:**

This paper proposes a new perspective of looking at backdoor attacks as features
rather than outliers. It states that backdoor attacks are impossible to detect
from natural features without structural information on training data
distribution. It also assumes that a backdoor attack is the strongest feature
and designs a defense framework based on this assumption.

**Summary Of The Review:**

This paper defines features in a limited way and assumes backdoor attacks as
the strongest features without model learnability consideration. These seem to
be fundamental limitations.

---

> ### Author Response · Authors · 2022-11-17
> **Response to Reviewer 9GLS**
>
> - _Its definition seems to be oversimplified. The assumption it makes seems to be wrong. Evaluation is weak._
>
> The goal of this paper is to go from a simple set of definitions and assumptions that are realistic, and explore theoretically the data poisoning attack. We believe that the definitions are expressive enough to derive interesting theoretical results, without overcomplicating these definitions. The assumptions we make are also valid, since in many backdoor attacks, the trigger has a stronger effect than other features (assumption 1). Furthermore, datamodels could be used to approximate the expected margin, as validated extensively in the datamodeling paper (Ilyas et. al 2022).
>
> - _Its definitions of features and feature supports seem to be oversimplified... I am not convinced that an image in DNN can be analyzed this way. If so, some hard problems in DNN will become easy._
>
> An exact definition of a feature is hard. We instead keep the definition general, yet specific enough to derive interesting theoretical results. For example, think of a feature “dog”; in that case, it is fairly simple to assess whether an image contains a dog or not. In the context of backdoor attacks, this definition is well-defined, as it is equivalent to the question: does this image contain a (potentially hidden) trigger?
>
> - _Another fundamental issue of this work is that it assumes that backdoor attacks are the strongest features without considering the learning process.  This paper assumes that backdoors as stronger features than natural features originally occur in input samples. This is wrong. I want to point out that many datasets do have strong natural features. In backdoor research, this is known as the natural backdoor._
>
> We respectfully disagree with the reviewer. We agree that natural features can be “strong” as defined in our paper. We kindly refer the reviewer to Figure 4 in our paper, where we show that in ImageNet a simple picture of a tennis ball is an effective pre-existing/”natural” backdoor. The reference (Liu et al., 2019) points out a similar phenomenon: attempting to invert backdoor triggers in a “clean” dataset recovers deer antlers -- a strong feature. The key idea behind our reformulation is that *if the backdoor trigger is not the strongest feature, then there is no reason to believe we should be able to find it*. For example, suppose we plant an artificial backdoor trigger in ImageNet, which leads to a backdoor attack that is less effective than the naturally-occuring “tennis ball” attack. An optimal backdoor-finding procedure should flag tennis balls, not the artificial triggers -- hence our assumption. In short, if the backdoor trigger is not the strongest feature, it is not the most *effective* backdoor, and hence it is not a well-defined problem to flag it over the stronger naturally-occurring backdoor. To reiterate, our central claim is that the problem of finding/flagging/neutralizing backdoor attacks is hopeless in its full generality. We then propose a minimal assumption that makes the problem well-defined and introduce a simple algorithm to flag backdoors that provably detects backdoors when the minimal assumptions hold.
>
> Liu, Yingqi, et al. "Abs: Scanning neural networks for back-doors by artificial brain stimulation." Proceedings of the 2019 ACM SIGSAC Conference on Computer and Communications Security. 2019.
>
> - _Another line of work is trying to develop a robust learning method, e.g., ABL by Li et al. NeurIPS 2021, in the face of poisoning data. They also prove that the learning process can pick up more desirable features in training to produce more robust models. This also shows that the assumption of this work does not hold in practice._
>
> The reference that the reviewer brings up argues that samples with backdoor triggers are often learned earlier in the training process, and provides a heuristic to defend against backdoor attacks that follow this pattern. We do not see how this is relevant to either of our two assumptions. We kindly ask the reviewer to elaborate how this might relate to either feature strength, or accuracy of datamodels.
>
> - _This paper defines features in a limited way and assumes backdoor attacks as the strongest features without model learnability consideration. These seem to be fundamental limitations._
>
> It is definitely possible to have a natural feature stronger than a crafted backdoor. Our framework surfaces a subpopulation corresponding to the strongest feature, and this feature would be more effective as a backdoor for a poisoning attack compared to the artificially-crafted trigger. The goal of finding the inserted trigger is then equivalent to finding the “strongest artificially-crafted backdoor”, which is not well-defined.

---

### Official Review · Reviewer_RpQm · 2022-10-25

**Confidence:** 4
**Correctness:** 3
**Technical Novelty And Significance:** 3
**Empirical Novelty And Significance:** 2
**Recommendation:** 5

**Clarity, Quality, Novelty And Reproducibility:**

The novelty is in the new perspective on (the most common form of) backdoor attacks as well as using the “data model” tool from the recent work of Ilyas et al. for their heuristic defense, which looks like a nice application of this tool.

The theoretical section is too dense and many parts of it seem not fully baked to me.

Some comments about the writing that might be helpful to the authors:
You keep mentioning the “indistinguishability” claim, but it is never formalized.
At the minimum, you could discuss the connection to “clean label” attacks. See here for the literature:
https://arxiv.org/pdf/2012.10544.pdf

You mention the “robust statistics” as the default source of literature on poisoning attacks. This is not true. You are working in the supervised setting, and there is already a large body of work that is much more relevant to your setting. For example, look up the words ``malicious noise” or “nasty noise” from the classical works Valiant and others.

I do not understand the claims about why the claim about the “slopes” in Figure 2 holds. I think the slopes are always there and exist, but it can be the case that in Def 3, the event that you condition on has zero probability. So something does not match.


**Strength And Weaknesses:**

The strength is to give a new perspective on backdoor attacks. The perspective is intuitive and might lead to useful defenses.

Weakness: There is basically no real theoretical result in this paper. Everything is based on (numerous) assumptions, which expectedly confirms that the defense works, but there are so many assumptions that are thrown without much deep justifications.

One can argue that the theoretical part would not matter as long as this defense truly works in practice. I agree with that sentiment, but here comes the second (main) weakness:

The paper only studies previous attacks, while clearly an adversary might have this defense in mind when attacking. This has happened over and over in adversarial learning tha defenses carefully study previous attacks, while attacks are defined adaptively. At the bare minimum, the paper should try to experiment with attackers who try to, e.g., keep the margin and sensitivity bounds of their adversarially added examples below a certain threshold to fool this defense. That is why I cannot support this paper for the top venue of ICLR if it does not follow this (by now well known) standard to judge defenses.

Another weakness is that: the paper really focuses on a specific type of backdoor attacks. In general, a backdoor attack can do *any* form of poisoning as long as (1) it has few poison points (2) it can control the prediction by a small amount of perturbation. This paper, on the other hand, focuses only on the “most common” way of doing the backdoor, which is by adding a feature to a bunch of examples.


**Summary Of The Paper:**

The paper gives a new perspective on backdoor attacks and how to identify and defend them. In back-door attacks, an adversary first performs a poisoning attack, usually by adding a specific pattern to examples that are then labeled carefully. Then, at the test time, the adversary can control the predicted label by minimally modifying the example (e.g., by again adding the pattern).

The paper focuses on the most common way of doing the backdoors (i.e., by adding patterns) and makes the following contributions:

1. Aiming to identify what backdoor attacks are: The paper starts by making the case that “feature adding” attacks are, in general, hard to identify as those features look like other features. Then the paper goes ahead and (somehow contradicts this thesis and) says that such adversarial features have special forms of features that could be identified as follows. The paper defines a notion of “senstiivty” for test examples with respect to a specific feature (where a feature is generally thought of as a boolean function over the training set) for a given data set and an integer k. Intuitively, this captures how much the answer to that test flips, when we go from “image” set of size k to “image” set of size k+1, where “image set” is the set of examples from the training set that have the feature.

2. Preventing the backdoor attacks: once we accept the thesis of item (1), that backdoor attacks will be identifiable by finding the “feature” with maximum “strength”. Then, the paper shows a heuristic that aims to identify such feature by (heavily) relying on the recent work of “data models” (from 2022) that allows one to predict the change of labels through a vector in R^|S| where S is the training set. This vector, once used in an inner product with the characteristic function of a subset, will give the result of training on that subset. Using this strong tool (which is conjectured to work well for neural nets) the paper defines an optimization task that heuristically solves to identify a feature/subset that is adversarial.

3. Finally, the paper performs some experiments to see how well their defense against two *previously known* attacks.


**Summary Of The Review:**

This paper gives a new interesting perspective on backdoor attacks. It then uses this perspective to come up with a new heuristic defense. The theory part is a bit sketchy and the experimental part does not aim for studying attacks that have this defense in mind.

---

> ### Author Response · Authors · 2022-11-17
> **Response to Reviewer RpQm**
>
> - _There is basically no real theoretical result in this paper. Everything is based on (numerous) assumptions, which expectedly confirms that the defense works, but there are so many assumptions that are thrown without much deep justifications._
>
> Every theoretical work requires a set of assumptions. In this work, we only have two intuitive assumptions, which we leverage to study the effect of the poison on the model’s predictions. To the best of our knowledge, this is the first theoretical work that studies end-to-end backdoor attacks, and leads to a theory-inspired defense.
>
> - _One can argue that the theoretical part would not matter as long as this defense truly works in practice._
>
> We provide a general and flexible theoretical framework that could be leveraged to study backdoor attacks. More assumptions can be made about specific attacks to come up with a corresponding defense. We hope this work will inspire future efforts in studying other types of attacks.
>
> - _the paper should try to experiment with attackers who try to, e.g., keep the margin and sensitivity bounds of their adversarially added examples below a certain threshold to fool this defense._
>
> If you design a poison that keeps the margin constant, then the trigger will have no effect on the model’s predictions, and the attack will no longer be effective.
>
> - _Another weakness is that: the paper really focuses on a specific type of backdoor attacks. In general, a backdoor attack can do any form of poisoning as long as (1) it has few poison points (2) it can control the prediction by a small amount of perturbation. This paper, on the other hand, focuses only on the “most common” way of doing the backdoor, which is by adding a feature to a bunch of examples._
>
> The literature of data poisoning is vast, and our goal in this paper is to present a new perspective to study theoretically the poisoning attack, then derive a practical algorithm to counter the attack. The hope is to motivate future work to study more threat models within this framework, in order to build better defenses.
>
> - _I do not understand the claims about why the claim about the “slopes” in Figure 2 holds. I think the slopes are always there and exist, but it can be the case that in Def 3, the event that you condition on has zero probability. So something does not match._
>
> Indeed, the slopes are always there and they represent how the margins change when the number of poisoned samples in the dataset changes. We leverage the values of these slopes to quantify the effect of adding more poisoned samples to the training set.

---

### Official Review · Reviewer_nR1F · 2022-11-01

**Confidence:** 4
**Correctness:** 3
**Technical Novelty And Significance:** 3
**Empirical Novelty And Significance:** 3
**Recommendation:** 5

**Clarity, Quality, Novelty And Reproducibility:**

The work was fairly clear, but seems to contain several typos. The work is novel as far as I know.

**Strength And Weaknesses:**

### Strengths:
1. I like how the authors began their work in a principled way by discussing difficulties with categorizing backdoor attacks.
2. The approach seems fairly intuitive and well motivated.

### Weaknesses:
1. I don't think the authors can claim contribution for presenting a "new perspective" on backdoor attacks as it is common knowledge that backdoor triggers can be discriminative features of the data. Also, I think this analysis applies more to the "clean label" setting - a point I would make clearer in the manuscript.
2. The work only deals with rudimentary backdoor attacks. It would have been nice to test on modern backdoor attacks - some of which even disguise the "trigger" e.g. [1,2].
3.  The empirical results seem lukewarm - compared to some older defenses, the proposed approach wins out, but often it is not the most successful defense.
4. There is a lot of clunkiness with the definitions/assumptions. For example:
    * You might be missing an $S'$ in the Definition 2 - $f(x;S')$ vs. $f(x;S)$
    * I don't like the presentation for Definition 1. If you're calling $\phi$ the feature $\phi:S \rightarrow \{0, 1\}$, then define $\textit{another}$ symbol to refer to the preimage of $\{1\}$. When you reuse $\phi$ to define $\phi(S) = \{ x | \overline{\phi(x)} = 1 \}$ Also, what is $\overline{\phi(x)\}$? Also what is feature $f$? Is this a typo?
    * Why have $\mathcal{S} = (X \times Y)^n$? This seems unnecessary. Just define $\mathcal{S}$ as $X \times Y$ for some input space $X$ and label space $Y$.

[1] Saha, Aniruddha, Akshayvarun Subramanya, and Hamed Pirsiavash. "Hidden trigger backdoor attacks." Proceedings of the AAAI conference on artificial intelligence. Vol. 34. No. 07. 2020.

[2] Souri, Hossein, et al. "Sleeper agent: Scalable hidden trigger backdoors for neural networks trained from scratch." arXiv preprint arXiv:2106.08970 (2021).

**Summary Of The Paper:**

The authors introduce a defense to backdoor attacks based on detecting "strong" features in training data, and removing backdoored training data by filtering along these lines.

**Summary Of The Review:**

The lukewarm results and less than ideal presentation makes me lean toward rejection.

---

> ### Author Response · Authors · 2022-11-17
> **Response to Reviewer nR1F**
>
> - _I don't think the authors can claim contribution for presenting a "new perspective" on backdoor attacks as it is common knowledge that backdoor triggers can be discriminative features of the data. Also, I think this analysis applies more to the "clean label" setting - a point I would make clearer in the manuscript._
>
> To the best of our knowledge, we have not found a work that formalizes the above. We would appreciate it if you can provide some reference.
>
> - _You might be missing an $S'$ in the Definition 2 - $f(x;S')$ vs. $f(x;S)$._
>
>  We thank the reviewer for spotting this typo. We will update this in upcoming versions of the manuscript.
>
> - _I don't like the presentation for Definition 1. If you're calling $\phi$ the feature $\phi:S→\{0,1\}$, then define another symbol to refer to the preimage of 1. When you reuse $\phi$ to define $\phi(S)=x|\phi(x)=1$. Also, what is $\phi(x)$? Also what is feature $f$? Is this a typo?_
>
> We thank the reviewer for spotting the notation overload of $\phi$. We will update this in future versions of the manuscript. $\bar \phi(x)$ is an unfortunate stylistic situation -- the defined word “support” on the line above is underlined
>
> - _Why have $S=(X\times Y)^n$ ? This seems unnecessary. Just define $S$ as $X \times Y$ for some input space $X$ and label space $Y$._
>
> $S$ is a fixed (training) set containing n examples. This fact is needed (and used) throughout the remainder of the section

---

> > ### Comment · Reviewer_nR1F · 2022-11-21
> > **Response**
> >
> > Thank you for your response.
> >
> > I appreciate the clarifications from the authors. With respect to the "new perspective", I'm not sure if I know of works that explicitly formalize the notion of backdoors as inducing discriminative features, but the understanding of this phenomenon can be found in several works, including [1,2].
> >
> > I'm interested if the authors were able to extend the work to other backdoor attacks that might be stronger?
> >
> > [1] Turner, Alexander, Dimitris Tsipras, and Aleksander Madry. "Clean-label backdoor attacks." (2018).
> > [2] Gao, Yansong, et al. "Backdoor attacks and countermeasures on deep learning: A comprehensive review." arXiv preprint arXiv:2007.10760 (2020).

---

### Official Review · Reviewer_kC2s · 2022-11-04

**Confidence:** 4
**Correctness:** 3
**Technical Novelty And Significance:** 2
**Empirical Novelty And Significance:** 2
**Recommendation:** 3

**Clarity, Quality, Novelty And Reproducibility:**

__Clarity:__

($+$) This paper has a well-organized structure and is concisely written.

($-$) The main formulation seems to be borrowed from (Ilyas et al., 2022) which is not properly referred to. For example, definition 2 (margin function) is the same as Eq. 7 (correct-class margin) in (Ilyas et al., 2022). As this metric is heuristically chosen by (Ilyas et al., 2022), I believe it should be cited.

($-$) Definitions are mostly not written formally in math but only with a line of words. It is helpful to have words to help the audience understand the concepts but they should only follow the mathematical formulations, not by themselves alone. (Ilyas et al., 2022) also used similar text definitions, but always accompanied with mathematical notations.

($-$) Some definitions/notations are confusing to me:
  - Informal Assumption 1: "We assume that adding a backdoor example ($x:\phi_p(x) = 1$) to the training set changes predictions on the backdoor examples  $\phi_p(S)$ more than adding an example $x : \phi_(x) = 1$ changes predictions on  $\phi_(S)$."
    - $\phi_p(S)$ is not formally defined. Is it possible that $\phi_(x)=\phi_p(x) \ \forall x\in\mathcal{X}$? In this case, does this statement still hold?
  - Definition 3:
    - I don't know if "$S'\sim\alpha S$" is a valid expression for "sampling a random α-fraction subset of S." $S'\sim S, |S'|=\alpha |S|$ could be a better one.
    - A definition of $k$ should be given near Definition 3.
  - Assumption 1:
    - I didn't get what $p$ is given the definition $p:=|\phi_p(S)|=|\phi(S)|$ . Why is $|\phi_p(S)|=|\phi(S)|$? Does it mean that a feature $\phi$ is considered in this assumption if and only if $|\phi_p(S)|=|\phi(S)|$?
    - $S_p$ is defined but not used in this assumption.
    - $\delta$ is not defined.

($\cdot$) I suggest adding a pseudocode for the algorithm.

__Quality__:

($\cdot$) I have a few questions regarding how the formal assumption, which is claimed as one of the contributions of this paper, is derived.
  * In the context of data poisoning, I don't know how I should understand the margin. If I understood the definition of margin correctly, it should be that the higher the margin, the better job the model is doing, right? Then why in Figure 2 do we see the average margin of the poisoned image increase when there are more poisoned samples? Can the authors help provide some intuitions about this increase?
  * Also for Figure 2, why should we look at the change in the margin of clean images when the number of poison images increases?
  * Why does Assumption 1 _sum_ sensitivity over the supports? I didn't follow how we arrive at Assumption 1 from Definition 3. It would be helpful if the authors can explain more.

($-$) Backdoor attacks and defenses considered in the experiments are limited and outdated. The model and the dataset options are also very limited: the only model used in the experiments is ResNet-9 and the only dataset tested is CIFAR-10. Experiments on more and larger models and datasets are needed.

__Novelty__:

($+$) Defending against backdoor attacks seems to be a novel usage of datamodels.

($-$) The empirical results are not significant compared to existing defenses like ISPL. The theoretical contribution is also marginally novel.

__Reproducibility__:

($+$) Very detailed information is provided for reproducing the results (though the code is not attached).

**Strength And Weaknesses:**

__Strength__:

* This paper is well-structured with all implementation details provided.
* Intuitition of theoritical claims is always given to help the audience understand the concepts.
* The idea of using datamodel for defending backdoor attacks is novel, and the proposed algorithm is very simple after the problem is provably reduced to a maximum-sum sub-matrix problem.

__Weaknesses__:

* Basic notions should be better defined and explained in the data poisoning setting.
* Connections between definitions and assumptions in Section 3 are weak and hard to follow.
* As a result of the bullet point above, the theoritical intuitions cannot convincably motivate the algorithm.
* The experiment section lacks more up-to-date attacks and defenses and more larger datasets and models. The results are also not significantly better than compared methods.

**Summary Of The Paper:**

This paper approached the data poisoning problem from a feature perspective. It argued that the backdoor is no more than the strongest feature in the dataset and thus can be detected based on its high sensitivity. The authors proposed using datamodels to approximate the sensitivity of features and tested the methods on dirty-label and clean-label attacks with either fixed pattern or m-way triggers. Compared to Inverse Self-Paced Learning (ISPL), Spectral Signatures and SPECTRE on CIFAR-10 and ResNet-9, the proposed algorithm successfully defended seven out of eight attacks. Still, it didn't outperform these existing methods on poisoned accuracy.

**Summary Of The Review:**

In my opinion, this is a work in progress rather than a complete work that meets the conference standard. The theoretical part is lacking in both novelty and clarity, and there should also be more experiments on the SOTA attacks and defenses with larger and more models and datasets to empirically validate the proposed algorithm. Therefore, I recommend rejection and encourage the authors to re-submit in the future after careful revision.

---

> ### Author Response · Authors · 2022-11-17
> **Response to Reviewer kC2s**
>
> - _(−) The main formulation seems to be borrowed from (Ilyas et al., 2022) which is not properly referred to. For example, definition 2 (margin function) is the same as Eq. 7 (correct-class margin) in (Ilyas et al., 2022). As this metric is heuristically chosen by (Ilyas et al., 2022), I believe it should be cited._
>
> The definition of the margin is a basic concept in machine learning, where we measure the signed distance of a point to the decision boundary. The work of Ilyas et al., which we use and cite extensively, studies the choice of target function for the datamodels.
>
> - _(−) Definitions are mostly not written formally in math but only with a line of words. It is helpful to have words to help the audience understand the concepts but they should only follow the mathematical formulations, not by themselves alone. (Ilyas et al., 2022) also used similar text definitions, but always accompanied with mathematical notations._
>
> We provide formal definitions of every term in section 3. e.g. for the definition of “feature” we first give a (purposefully broad) mathematical definition, followed by intuition. For a task with input space $\mathcal{X}$ and label space $\mathcal{Y}$, and for a fixed dataset $S \in (\mathcal{X} \times \mathcal{Y})^n$, a feature is simply a function $f \in \mathcal{X} \to \{0, 1\}$. For example, $f$ might map from an image $x \in \mathcal{X}$ to whether the image contains "dog".
>
> - _$\phi_p(S)$ is not formally defined. Is it possible that $\phi(x)=\phi_p(x) ∀x∈\mathcal{X}$? In this case, does this statement still hold?_
>
> $\phi_p(S)$ is the feature support of the feature $\phi_p$, as defined in Definition 1, which we believe is clearly stated in the assumption. It is indeed possible for $\phi_p$ and $\phi$ to be perfectly correlated. This is a degenerate case, for which datamodels will not be able to differentiate between the two features. Thus it is not critical to assume that Assumption 1 holds in this scenario.
>
> - _I don't know if "$S' \sim \alpha S$" is a valid expression for "sampling a random $\alpha$-fraction subset of $S$." $S'\sim S,|S'|=\alpha |S|$ could be a better one._
>
> We use that notation for brevity. We will update in upcoming versions.
>
> - _A definition of $k$ should be given near Definition 3._
>
> $k$ is a variable in def. 3
>
> - _I didn't get what $p$ is given the definition $p:=|\phi_p(S)|=|\phi(S)|$. Why is $|\phi_p(S)|=|\phi(S)|$? Does it mean that a feature $\phi$ is considered in this assumption if and only if $|\phi_p(S)|=|\phi(S)|$?_
>
> The reviewer’s interpretation is correct. We acknowledge the writing is confusing and we will restate this part of the assumption in upcoming versions of the manuscript.
>
> - _$S_p$ is defined but not used in this assumption._
>
> This is indeed a typo, we thank the reviewer for spotting this.
>
> - _$\delta$ is not defined._
>
> We thank the reviewer for spotting this, we will add a quantifier (for some absolute constant $\delta >0$)  in the body of the assumption.
>
> - _In the context of data poisoning, I don't know how I should understand the margin. If I understood the definition of margin correctly, it should be that the higher the margin, the better job the model is doing, right? Then why in Figure 2 do we see the average margin of the poisoned image increase when there are more poisoned samples? Can the authors help provide some intuitions about this increase?_
>
> Correct, the higher the margin, the more confident the model is that a particular image $x$ has the label $y$. In the context of a backdoor attack, the margin is with respect to the image label, which could be the correct class (clean-label attack) or an incorrect class (dirty-label attack). When there are more poisoned images in the dataset containing the trigger t and having a poisoned label $y_{p}$, the more the model is confident that an image $x$ with the trigger $t$ has the label $y_{p}$, and consequently the higher the margin.
>
> - _Also for Figure 2, why should we look at the change in the margin of clean images when the number of poison images increases?_
>
> The goal is to illustrate that the number of poisoned images in the dataset affects the margins of the model for poisoned samples, and doesn’t affect the margin of the clean samples.
>
> - _Why does Assumption 1 sum sensitivity over the supports? I didn't follow how we arrive at Assumption 1 from Definition 3. It would be helpful if the authors can explain more._
>
> We define in our paper the support of a feature f as the training samples that contain this feature. We also quantify the sensitivity of a model towards a particular feature by measuring how the margin of the model changes when more samples with that feature are included in the dataset. Given that definition, we make an assumption that the feature corresponding to the poison trigger has a stronger effect on the model compared to another feature in the dataset. This translates into a higher sensitivity towards poison features compared to other features.

---

> > ### Comment · Reviewer_RpQm · 2022-11-24
> > **Thanks for answers**
> >
> > Thanks for your points. I wanted to add some points, in case they are helpful:
> >
> > - there are more assumptions than 1 and 2 (even thought these are already very strong assumptions need better justification and clarification). for example, the condition on $\delta^*$ in Lemma 2 is an assumption itself.
> >
> > On a related note: I agree that every theory result comes with some assumptions, but it is always about how natural/justified they are. Simply saying that they always exist ignores the main issue.
> >
> > Going back to the main issue:
> > I think simply saying that you initiate a defense approach is not good enough maybe for an ICLR paper. Defense mechanism should at least try to be "adaptively secure". This is how e.g., security has always been defined in Cryptography and doing the same approach in Adversarial Learning is necessary too. It is not enough to only study attacks that are completely unaware of your defense. Namely, you should at least *try* to come up with defenses that are "adaptively secure" (again in the sense that attacks knowing your defense will still fail).
> >
> > I mentioned that "it can be the case that in Def 3, the event that you condition on has zero probability" but as I understood, I think your response did not clarify what happens in this case.
> >
> > thanks again.

---

### Author Response · Authors · 2022-11-17
**General comment to the reviewers**

We thank the reviewers for their helpful feedback. Following is a general reply to the reviewers’ questions.

We would like to emphasize that the goal of this paper is to present a new framework for thinking about data poisoning. In particular, we seek to quantify – from a data perspective – the effect of backdoor attacks on a models’ behavior. In particular, we show that even in a simplified setting, the problem of detecting backdoored images is not well-posed in its full generality. We propose a set of minimal assumptions. Given these assumptions, we propose a theory-inspired practical algorithm that leverages datamodels in order to detect a subpopulation with a strong effect on a model’s predictions. To validate our theoretical framework, we execute several backdoor attacks, and show the effectiveness of our defense when our assumptions hold. We also consider experiments where some of our assumptions do not hold to show when our algorithm fails. The literature of backdoor attacks is very vast, and evaluating against all emerging algorithms is better tailored towards a benchmark paper, or a survey. Our selection of attacks was based on showing a proof-of-concept of our theoretical framework. We hope to motivate future work to investigate other threat models in data poisoning using our framework in order to build a repertoire of theory-inspired algorithms.

---

### Decision · Program_Chairs · 2023-01-20

**Decision:**

Reject

**Justification For Why Not Higher Score:**

N/A

**Justification For Why Not Lower Score:**

N/A

**Metareview: Summary, Strengths And Weaknesses:**

Under the assumption that a backdoor attack corresponds to the strongest feature in the training data, this paper develops a new framework for detecting backdoor attacks. The proposed algorithm is evaluated both theoretically and experimentally.

Strengths:
a new perspective on backdoor attacks
The paper is well-motivated

Weaknesses:
Lack experiments on SOTA attacks
The analysis relies on too strong assumptions

Even after the author response, the concerns remain unresolved. Therefore, I have to recommend rejection.



**Summary Of Ac-Reviewer Meeting:**

N/A